# Physics-Constrained Comprehensive Optical Neural Networks

**Yanbing Liu[1,*]**
ybliu@bupt.edu.cn

**Jianwei Qin[2,*]**
qinjw3@sjtu.edu.cn

**Yan Liu[2]**
liu_yan@sjtu.edu.cn

**Xi Yue[1]**
yuexi@bupt.edu.cn

**Xun Liu[3]**
liuxun_laby@163.com

**Guoqing Wang[4]**
gqwang0420@hotmail.com

**Tianyu Li[4]**
cosmos.yu@hotmail.com

**Fangwei Ye[2,†]**
fangweiye@sjtu.edu.cn

**Wei Li[3,†]**
wei_li_bj@163.com

[1] Beijing University of Posts and Telecommunications, Beijing,100876,China
[2] Shanghai Jiao Tong University, Shanghai 200240, China
[3] Beijing Institute of Space Mechanics and Electricity,Beijing,100094,China
[4] University of Electronic Science and Technology of China, Chengdu, 611731, China

## Abstract

With the advantages of low latency, low power consumption, and high parallelism, optical neural networks (ONN) offer a promising solution for time-sensitive and resource-limited artificial intelligence applications. However, the performance of the ONN model is often diminished by the gap between the ideal simulated system and the actual physical system. To bridge the gap, this work conducts extensive experiments to investigate systematic errors in the optical physical system within the context of image classification tasks. Through our investigation, two quantifiable errors—light source instability and exposure time mismatches—significantly impact the prediction performance of ONN. To address these systematic errors, a physics-constrained ONN learning framework is constructed, including a well-designed loss function to mitigate the effect of light fluctuations, a CCD adjustment strategy to alleviate the effects of exposure time mismatches and a 'physics-prior-based' error compensation network to manage other systematic errors, ensuring consistent light intensity across experimental results and simulations. In our experiments, the proposed method achieved a test classification accuracy of 96.5% on the MNIST dataset, a substantial improvement over the 61.6% achieved with the original ONN. For the more challenging QuickDraw16 and Fashion MNIST datasets, experimental accuracy improved from 63.0 to 85.7% and from 56.2% to 77.5%, respectively. Moreover, the comparison results further demonstrate the effectiveness of the proposed physics-constrained ONN learning framework over state-of-the-art ONN approaches. This lays the groundwork for more robust and precise optical computing applications.

## 1   Introduction

In recent years, optical neural networks (ONNs) have garnered significant research attention for inference tasks such as object detection and object classification[1, 2, 3, 4], attributed to their advantages

---

[*]Equal contribution
[†]Corresponding Author

38th Conference on Neural Information Processing Systems (NeurIPS 2024).

of low energy consumption, high transmission speed, and large information capacity[5, 6, 7]. To establish ONNs, they are typically simulated as Deep Neural Networks(DNNs)[8, 9, 10] on electronic devices and trained using the backpropagation algorithm[11, 12, 13], after which the trained model parameters are deployed to physical ONN systems (as shown in Fig 1(a)). Theoretically, ONNs can maintain prediction performance comparable to that of their simulated electronic counterparts. However, during experimental implementation, errors are inevitably introduced, unexpectedly reducing their prediction accuracy[14, 15].

Some measurable errors in the optical system, such as light field perturbations caused by the scattering of impurity particles in the environment and optical distortions due to lens aberrations, can be mitigated by explicitly modeling them and integrating these error models into electronic training process[16, 17] (as shown in Fig 1(b)), while other unmeasurable errors, such as ambient light effects, laser instability, and crosstalk between light fields, are difficult to correct through physical modeling, posing significant challenges in compensating for discrepancies between simulations and experiments[18, 19, 20].

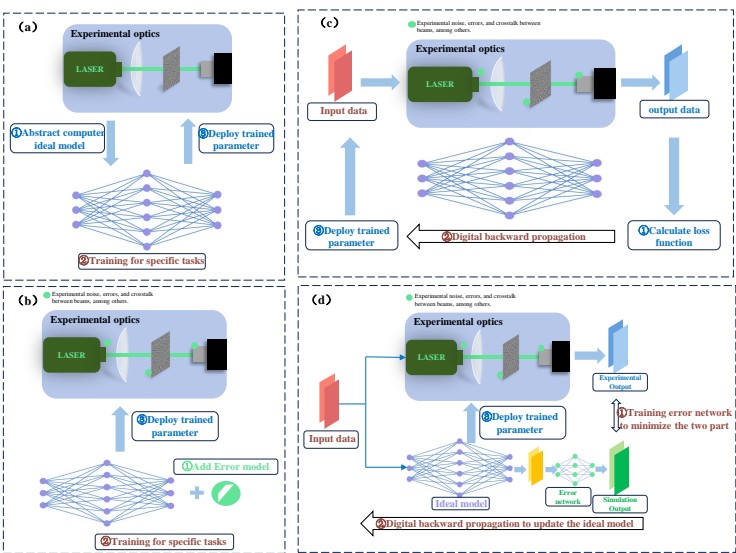

Figure 1: (a) Experimental schematic diagram of the ONN and corresponding simulated DNN on the electronic device. (b) Error compensation method based on the physical error model (c) Hybrid training method with pure digital DNN with error compensation functionality (d) Error compensation method with an ideal physical model and a digital error compensation DNN

The introduction of "hardware-in-the-loop" training techniques[21, 18, 22, 23] has opened up new possibilities for tackling the aforementioned challenges. Unlike in silico training, this hybrid training method incorporates the actual ONN physical response in each update loop, to mitigate the impact of system errors[24], as shown in Fig 1(c). This approach allows the simulated electronic neural network to capture the physical dynamic changes of the input light field along its propagation path, including the propagation process of the light field itself, the influences of various error sources and disturbances encountered. However, the simulated electronic neural network, especially for the primarily linear ONNs, might be too simplistic to learn the complex input-output light field mapping relationship, including light propagation, various errors, and the coupling between light fields.

Therefore, error compensation neural networks are proposed to better simulate the physical optical system [25], as illustrated in Fig 1(d). The addition of the error compensation neural network in the hybrid training scheme is expected to bridge the gap between simulation and physical systems. However, due to the large optimization space and the lack of physical constraints in the simulation system, the training process can be slow and may even diverge.

Inspired by the concept of Physics-Informed Neural Networks (PINNs)[26, 27, 28, 29], which integrate physical information into the network for optimization, we investigate systematic errors in the optical physical system first and then propose a physics-constrained ONN learning framework for image recognition tasks. In our approach, critical physical information is quantitatively integrated

into a 'physics-prior-based' error compensation network[30, 31]. This narrows the search space and reduces the complexity of the required DNNs. In our experiments, the error compensation network converges rapidly and effectively minimizes the disparity between simulation results and actual observations, which significantly enhances the image recognition accuracy of experimental ONNs, leading to state-of-the-art (SOTA) performance on several datasets.

The main contributions of this work are as follows:

**(1)** To describe the transmission equations of complex optical systems, we combine quantifiable physical information with machine learning. Given known ideal physical models and key parameters, we use minimal data and lightweight neural networks, leveraging both physics-driven and data-driven approaches for rapid and precise modeling of complex optical systems.

**(2)**We informed the network of two significant quantitative errors—laser source instability and camera exposure mismatch—as physical prior information, greatly enhancing convergence efficiency.

**(3)**Under multiple physical constraints, our network can focus more efficiently on learning other unmeasurable system errors beyond the two aforementioned quantitative errors, such as crosstalk between beams, device imperfections, and alignment errors. Consequently, it achieves state-of-the-art results across multiple datasets.

## 2   Related Work

**Training base on the ideal optical 4f system**    The 4f-ONN is constructed with two Fourier lenses and a Spatial light modulator (SLM) in the focal plane, modulating the frequency spectrum of the light field, which allows for the automatic realization of optical convolution operations and makes it well-suited for optical neural networks[32, 33, 3, 34]. Taking the optical 4f system as an example, in the absence of errors, its ideal transformation process adheres to the Fresnel diffraction[35, 36] integral under ideal conditions[15]. The loss between the simulated output optical field image and the target optical field is backpropagated through gradient descent, affecting the frequency domain phase distribution of the 4f system. Through multiple training iterations on the electronic device, an optimized spectral phase distribution is obtained. This phase distribution is then loaded onto the focal plane of the 4f system via the SLM, enabling image classification tasks.

**Fitting errors based on the physical model.**    [17, 16] aim to model the primary disturbances and errors in optical systems, integrating the disturbance model appropriately into the ideal simulated physical transformation model as an accurate description of the optical system. While this approach enhances the robustness of the model against certain interferences, it only considers one or a few disturbances existing in the physical system, thus unable to account for all disturbances, unmeasurable quantities, and their coupling and crosstalk within the model. Therefore, its description of the physical system model is not yet precise enough.

**ONNs auto-learning**    [24, 18] aims to describe real optical systems through autonomous learning using ONNs. By feeding input and output signals obtained from experiments into the ONNs, the ONNs learn the functionality of the optical system in a data-driven manner. Ultimately, this enables the network to accurately reflect the functionality of the actual optical system. However, learning the input-output relationships in experiments might be overly complex for optical neural networks, particularly for simple linear optical neural networks.

**Error compensation network**    [37, 25] integrates ideal physical models with deep neural networks (DNNs) in optical system modeling to compensate system errors and accelerate convergence speed. However, the lack of physical parameters and physics-constrained loss functions requires the network to learn this information from more data during training. Consequently, it necessitates the design of more complex DNN architectures and the acquisition of additional experimental data to support the training process.In contrast, through the integration of physical information into the simulated model, our method is able to achieve high prediction performance using a relatively simple error compensation network.

## 3  Method

### 3.1  Introducing quantifiable physical information into optical systems

Introducing quantifiable physical information into optical systems is a key strategy for enhancing the accuracy and efficiency of simulations in image recognition applications. The relationship between the input and output signals of the optical system in experiments can be represented by a simplified model, as depicted in Fig 2(a):

$$g(u) = f(u) + \Delta f_{\text{dev}}(u) + \Delta f_{\text{jit}}(u) + ... + \eta(u) \tag{1}$$

where $f(u)$ denotes the ideal transformation process, $\Delta f_{\text{dev}}$ indicates deviations caused by imperfections in optical devices, $\Delta f_{\text{jit}}$ represents deviations due to laser jitter, and $\eta$ encompasses deviations caused by unmeasurable quantities in the system.

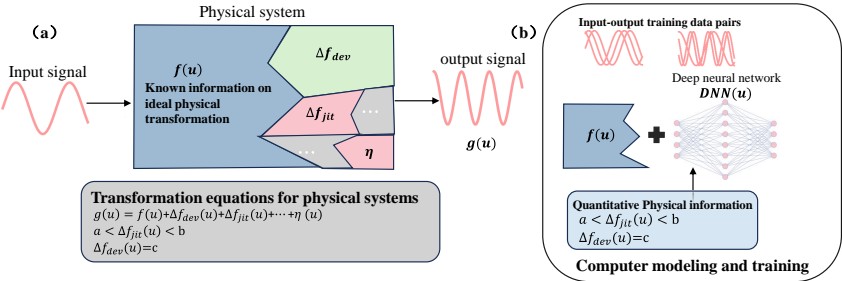

Figure 2: Schematic diagram illustrating the integration of physical information to reduce experimental errors in the precise simulation of physical systems.

Accurate simulation of the actual optical signal transmission process necessitates not only simulating the ideal transformation of the light path but also compensating for system errors, as illustrated in Fig 2(b). Here, $f(u)$ symbolizes the ideal transformation used in the simulation, and DNN is employed as the neural network for error compensation, with $\Delta f_{\text{dev}}$ and $\Delta f_{\text{jit}}$ representing quantifiable system errors. Methods such as the Finite Difference Time Domain[38, 39, 40] (FDTD) for solving the Helmholtz equation or the paraxial Schrödinger equation using the split-step Fourier transform method[41, 42, 43] are utilized to effectively simulate the ideal transformation $f(u)$ of the input light field. The primary role of the DNN is to learn from the difference between the output of $f(u)$ and the experimental output, focusing on experimental errors beyond $f(u)$ and their coupling effects—a complex task. To reduce the complexity of the DNN and ensure the convergence of our model, quantifiable physical information, such as the range of laser jitter and the grayscale value range of images received by cameras, is incorporated into the DNN. Through carefully designed loss functions[44, 45, 46] and by adjusting the overall data bias,this quantifiable physical information is integrated into the DNN model, thereby reducing its complexity and significantly increasing convergence speed.

### 3.2  Architecture of optical neural networks and measurement of quantifiable experimental errors

We employ the method described above to precisely simulate the light propagation process within an image-classification optical neural network, where the experimental errors contribute to a reduction in classification accuracy. The experimental architecture is illustrated in Fig 3(a), we established an experimental framework for an optical neural network with error compensation capabilities. Detailed descriptions of the experimental setup can be found in the supplementary materials. Using the Fresnel diffraction integral, we can derive the ideal transformation process within this optical system (i.e., the function $f$ in Equation.1). This setup allows for precise control and manipulation of the light field, facilitating detailed investigations into the dynamics of image processing within optical neural networks.

$$E_{out} = \frac{\psi_{sph}}{i\lambda f_1 f_2} \cdot [t(x, y) * e^{i\phi(x', y')}] \tag{2}$$

$$f = min\{|E_{out}|^2, I_{max}\} \tag{3}$$

where '*' denotes the convolution operation, $f_1$, $f_2$ are focal length of L1 and L2, respectively. $t(x, y)$ is the input image in DMD, loading in the form of a binary transmission function. $\phi(x', y')$ represents for the phase distribution on SLM and $\psi_{sph}$ is a spherical wave phase factor. Due to the inherent limitations of CCD which can only detect light intensity, and given that the maximum intensity that the CCD can read is $I_{max}$, the output light field must transform as dictated by Equation.3. After transforming the optical field of the input image using the optical 4f system and SLM, the output optical field is obtained. Taking the MNIST dataset as an example, the output optical field is divided into 10 equally sized regions for recognition and classification. The light intensity $I_i(i = 1, 2, ..., 10)$ in each region is measured to form the intensity sequence $[I_1, I_2, ..., I_{10}]$. The region with the highest light intensity corresponds to the classification result.

In this optical system, beyond the ideal transformation process $f$, some quantifiable physical quantities can be utilized to simplify the complexity of the error compensation network. Two primary quantifiable experimental errors include the instability of laser intensity and the consistency of light intensity between the output image and the simulated image. To quantitatively analyze and mitigate these errors, we measured the instability of laser intensity and the impact of camera exposure time on the experimental output image.

### 3.2.1 Quantitative compensation of laser intensity instability

For the instability of laser intensity $\Delta f_{jit}$, we used a CCD to measure the overall grayscale values within a fixed area to assess the stability of the laser. Over 700 minutes, the variation of pixel values changed randomly, as depicted in Fig 3(b). To incorporate errors caused by intensity instability into our error compensation network, A well-designed loss function is used to increase the gap between the maximum and second maximum intensity values within the classification area. The designed loss function is shown below:

$$Loss = ReLu\{W_{Gap} - |I_{max} - I_{2ndmax}|\} + \sum_{i=1}^{N} y_i log(\hat{y_i}) \tag{4}$$

Here, the $W_{Gap}$ denotes the light intensity gap, $N$ denotes the total number of classes, $y$ represents a one-hot encoded vector that indicates the true class labels, with $y_i$ being the $i$-th element of the vector $y$. The term $\hat{y}$ corresponds to the network's output probabilities, which are typically derived through the application of a softmax function, with $\hat{y_i}$ representing the probability that the model assigns to the likelihood that the sample pertains to class $i$. In the experiment, the gap is configured to substantially exceed the range of instability variations, thereby mitigating the reduction in experimental accuracy attributable to laser instability. However, the value of $W_{Gap}$ cannot be set too high, as this would reduce the network's fitting ability, leading to decreased classification accuracy. Here, we have experimentally measured the impact of different $W_{Gap}$ values on experimental accuracy (as shown in the inset of Fig 3(b)). When $W_{Gap}$ is set to 10, the experimental accuracy is optimal. In this setting, the network still maintains good fitting ability, and the instability of laser intensity is also compensated. This approach ensures that despite fluctuations in laser output, the performance of the optical neural network remains robust, providing reliable and precise experimental outcomes.

### 3.2.2 Quantitative compensation of exposure time mismatches

For the noise caused by optical devices $\Delta f_{dev}$, experimentally, the primary and measurable error source comes from the intensity of light received by the CCD. The distribution of pixel values read by the CCD corresponds to the distribution of light intensity it receives. However, inherent biases in the CCD itself result in discrepancies between simulated and experimental pixel values, leading to reduced experimental accuracy. The CCD's pixel values depend on the exposure time and the intensity of the emitted laser light. To eliminate this error, we fixed the output laser intensity and adjusted the exposure time of CCD to measure the gap between the simulated and experimental CCD pixel values. For each exposure time, we loaded 1000 images from the MNIST dataset onto the DMD and calculated the difference in pixel values across 10 classification regions, as shown in Fig 3(c). In the ideal case, the error is completely eliminated ($\delta I = 0$, as indicated by the red bars). In actual experiments, we first coarsely adjust the exposure time to bring the average $\delta I$ close to zero (green and blue bars in Fig 3(c)), and then finely adjust the exposure time to minimize the variance of $\delta I$

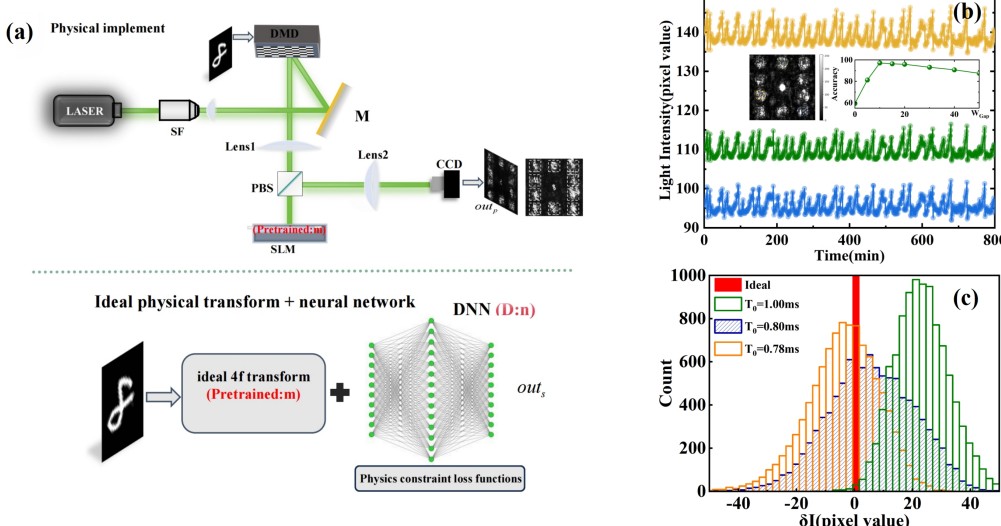

Figure 3: (a) Schematic of an image classification optical neural network with an error-compensating DNN incorporating quantitative physical information. (b) The random fluctuations of output light intensity, measured in three classification regions over 700 minutes. Inset: the experimental accuracy as a function of light intensity gap $W_{\text{Gap}}$. (c) The difference between simulated and experimental CCD reading values under various exposure times.

(orange bars in Fig 3(c)). This method effectively eliminates errors caused by the CCD and improves classification accuracy.

## 3.3 Training process of physics-prior-based error compensation network

In addition to the quantifiable system errors[47, 48, 49] previously discussed, several types of errors within optical systems are unmeasurable and can be categorized into fixed and coupling errors. Fixed errors arise from experimental deviations or mismatches, including the misalignment of the spatial light modulator, rotation of the input image, and diffraction effects due to lens apertures. Coupling errors, on the other hand, originate from the interaction between the light field and environmental impurities. As the phase map in the SLM is adjusted, while the light field is accordingly altered, the fixed errors remain unchanged, whereas the coupling errors vary. These experimental errors can affect the output light field of the 4f system, causing it to deviate from the output image in a perfect scenario, thereby reducing the recognition accuracy of ONN.

To mitigate the impact of such errors, a DNN network is used to model various environmental noises and experimental errors, aligning the ideal output image of the 4f system with the actual output image. The specific error compensation process is as below:

(i)*Pre-training*: Based on the Fresnel diffraction integral, train the simulational phase distribution $m_0$ of SLM in the ONN without considering any experimental errors and environmental noises.

(ii)*Error compensation*: Load the phase map $m_{k-1}$ onto the SLM to obtain the actual output image. Train the error compensation DNN $n_k$ to minimize the difference between the ideal and actual output image.

(iii)*Re-training*: Load DNN to compensate for the errors and retrain a new phase map $m_k$ of ONN to minimize the well-designed loss function (eq.4) in simulation. These two networks, ONN and DNN, are connected by residual connections.

(iv)*Iteration*: The coupling errors are altered when the phase map changes from $m_{k-1}$ to $m_k$. To compensate for that, repeat steps (ii) and (iii), iterating until the experimental classification accuracy is maximized.

Through such an iteration process, the error compensation network can model effectively the environmental noises and experimental errors, significantly improving the classification accuracy.

Incorporating quantifiable physical data, such as laser intensity fluctuations and camera exposure effects, directly into the network's architecture enables it to adapt more robustly to the inherent variability of optical systems.For more detailed information on the training process, please refer to the supplementary materials.

# 4 Result

## 4.1 Dataset

The MNIST (Modified National Institute of Standards and Technology)[50, 51] dataset consists of 70,000 28x28 pixel grayscale images of handwritten digits, widely used as a benchmark for training and testing image processing systems in machine learning and computer vision. The QuickDraw16 dataset, a subset of Google's Quick, Draw! project includes 16 categories of hand-drawn images that mimic natural variations in handwritten and sketched drawings, serving as a valuable resource for image classification and recognition tasks. The Fashion MNIST dataset[52], provided by Zalando, contains 70,000 28x28 pixel grayscale images categorized into 10 different fashion items, such as T-shirts, trousers, shoes, and bags. Designed as a direct replacement for the traditional MNIST dataset, Fashion MNIST is extensively used for research and evaluation in image classification tasks. During training, we used the entire training set. To evaluate accuracy, we randomly selected 1,000 images from the test sets of each of the three datasets to assess experimental accuracy.

## 4.2 Error compensation network without quantifiable physical information

In comparison to our approach of incorporating quantifiable physical information into the error compensation network, we also conducted an experiment where a DNN was trained to model the system errors without introducing any quantifiable physical data. As illustrated in Fig 4(a1), the network trained only on the ideal transformation process $f$ of the light field achieved an experimental accuracy of only 61.6%. This lower accuracy is due to several factors, including the instability of the laser $\Delta f_{jit}$, mismatches in exposure time $\Delta f_{dev}$, and various unmeasurable errors $\eta$, which resulted in significant deviations between the output light field images in experiments and simulations, as shown in Fig 4(a2-a3). The accuracy of image recognition depends on the distribution of light intensity across ten regions within the output image; thus, the experimental accuracy was significantly reduced.

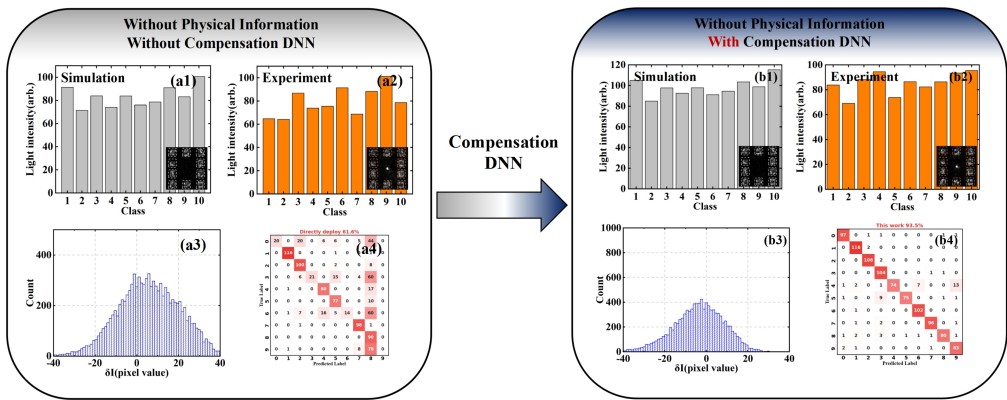

Figure 4: The simulation and experiment result with and without compensation DNN on MNIST dataset, maintaining without quantifiable physical information. (a1,b1) The simulational light intensity in ten classification regions is depicted in the inset. (a2,b2)The experimental light intensity in ten classification regions is depicted in the inset. (a3,b3) Histogram of light intensity difference $\delta I$ between simulation and experiment. (a4,b4)Experimental confusion matrix.

After incorporating an error compensation network without quantifiable physical information, the discrepancies in light intensity distribution between experiments and simulations were mitigated, as shown in Fig 4(b4), boosting the experimental accuracy to 93.5%. As indicated in Fig 4(b3), the role of the error compensation DNN at this point was twofold: to modulate the overall light intensity and to adjust the local distribution of light intensity, thereby narrowing the gap between

simulation and experiment. However, as observed in Fig 4(b1-b2), the modulation resulted in a small difference between the highest and second-highest light intensities, reducing the robustness of the neural network.

## 4.3 Error compensation network with quantifiable physical information

By integrating quantifiable physical information, we effectively compensated for system errors caused by laser instability $\Delta f_{jit}$, and mismatches in exposure time, represented as $\Delta f_{dev}$. Consequently, the DNN now primarily focuses on compensating for unmeasurable errors $\eta$, which are inherently more challenging to predict and correct. This approach has significantly improved the convergence speed and accuracy of the error compensation network. For the MNIST dataset, we mitigated the error $\Delta f_{dev}$ by adjusting the exposure of the CCD, thus ensuring that the overall light intensity between the simulation and experiment was consistent. This adjustment allowed the DNN to focus solely on modulating the local distribution of light intensity, thereby simplifying its complexity. As shown in Fig 4.3(a4), even without an error compensation network, the classification accuracy increased to 83.7% due to the consistency of the overall light intensity distribution.

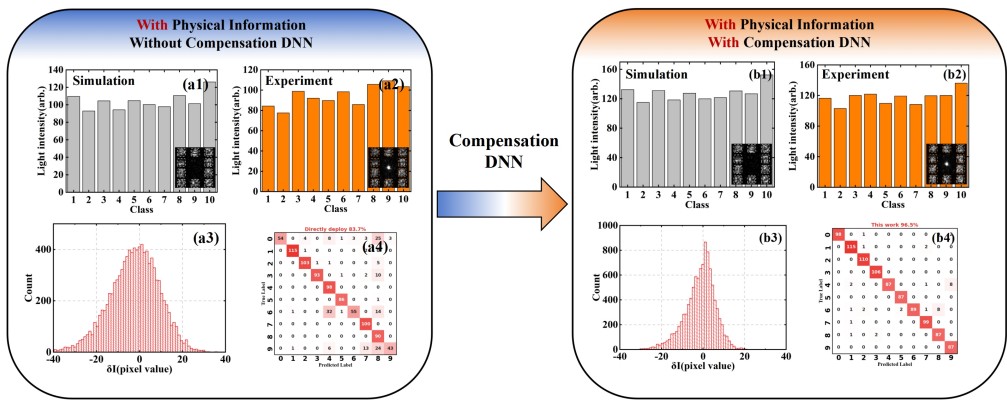

Figure 5: The simulation and experiment result with and without compensation DNN on MNIST dataset, maintaining with quantifiable physical information. (a1,b1) The simulational light intensity in ten classification regions is depicted in the inset. (a2,b2)The experimental light intensity in ten classification regions is depicted in the inset. (a3,b3) Histogram of light intensity difference $\delta I$ between simulation and experiment. (a4,b4)Experimental confusion matrix.

Additionally, a well-designed loss function was used to address the instability of laser intensity, $\Delta f_{jit}$, ensuring that the difference in light intensity between the maximum and the second maximum values within the classification regions was significantly greater than the range of fluctuations in laser intensity. After training, this error compensation network, enhanced with quantifiable physical information, increased the classification accuracy of the MNIST dataset to 96.5%.

## 4.4 Further experiment in different datasets

For the more complex Quickdraw16 dataset and FMNIST dataset, applying the same method of adjusting the CCD exposure to ensure consistency in overall light intensity, and using a well-designed loss function to address the instability of laser intensity significantly enhanced the recognition accuracy. The classification accuracy for the Quickdraw16 dataset increased from 63.0% to 85.7% through these adjustments. Similarly, the classification accuracy for the FMNIST dataset improved from 56.2% to 77.3% through these adjustments. The analysis of the convergence speed of the error compensation network can be found in the supplementary materials.Our approach can also handle more challenging datasets like CIFAR-10. To do this, we simply replace the dataset loaded onto the DMD with CIFAR-10. The initial classification accuracy with CIFAR-10 is 30%, which improves to 57% after optimization using our method. However, since current research[25, 37] on spatial 2D light-based ONNs predominantly uses the datasets mentioned in the paper, we do not present a detailed and specific demonstration of the results of CIFAR-10 in this paper due to the lack of comparative benchmarks.

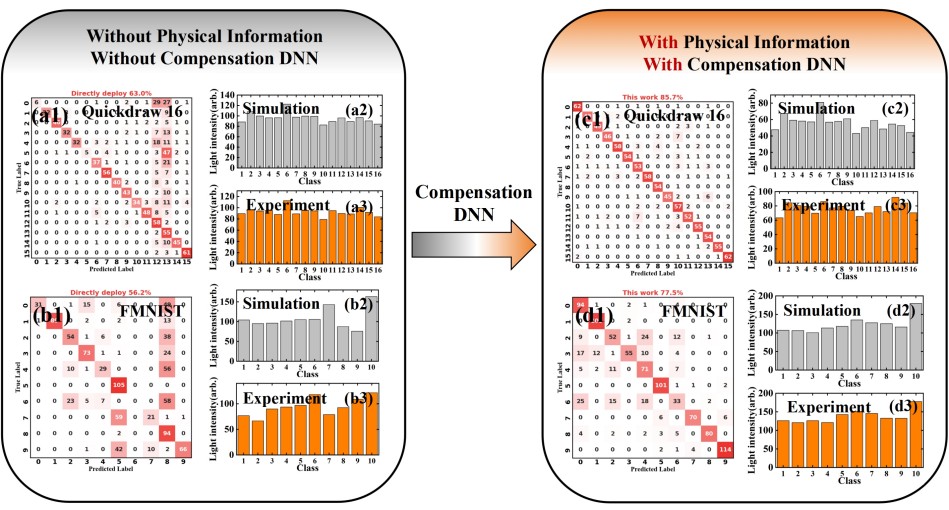

Figure 6: (a1) Experimental confusion matrix in Quickdraw 16 dataset without physical information and compensation DNN. (b1) Experimental confusion matrix in FMNIST dataset without physical information and compensation DNN. (c1)Experimental confusion matrix in Quickdraw 16 dataset with physical information and compensation DNN. (d1)Experimental confusion matrix in FMNIST dataset with physical information and compensation DNN.(a2,b2,c2,d2) The simulational light intensity in classification regions. (a3,b3,c3,d3) The experimental light intensity in classification regions.

## 4.5   Accuracy comparison with other error compensation network

We have compared our work with other studies and found that our experimental classification accuracy on the MNIST dataset aligns closely with the results from Tsinghua University[25] and those published in Nature magazine; on the QuickDraw16 dataset, we achieved an experimental classification accuracy of 85%. These results indicate that, unlike models that do not incorporate physical information, our approach of quantitatively introducing physical information has enabled our optical-DNN to be fed with known measurable experimental perturbations and errors. This allows the DNN that follows the ideal model to learn more about unknown perturbations, errors, and imperfections in optical devices, and their interdependencies. Consequently, compared to neural networks that do not incorporate physical information quantitatively, our optical-DNN(optica-Deep Neural Network) is able to more accurately represent real experimental systems, enhancing the consistency between simulation and experimental results, and further improving experimental accuracy. Additionally, our error compensation network is very lightweight, with approximately 5,000 learnable parameters(For more details, please refer to the supplementary materials). Therefore, the introduction of quantitative physical information into neural networks is crucial, especially when dealing with complex datasets that require highly precise simulations of actual physical processes.

| Architecture | Hybrid CNN[14] | DAT[25] | | PAT[37] | | This work Qualitative | | **This work Quantifiable** | |
|---|---|---|---|---|---|---|---|---|---|
| | | Directly deployed | optimize | Directly deployed | optimize | Directly deployed | optimize | Directly deployed | optimize |
| **MNIST** | | 24.9% | 92.4% | 24.9% | 61.4% | 61.6% | 93.5% | 83.7% | 96.5% |
| **FMNIST** | | 8.4% | 77.3% | | | | | 56.2% | 77.5% |
| **Quickdraw16** | 72.0% | | | | | | | 63.0% | 85.7% |

Table 1: The accuracy comparison between other error compensation network architecture in various datasets.

## 5   Conclusion

In this study, we explore the mapping relationship between input and output of a physical ONN system, which can be modeled as the sum of ideal transformation process, measurable errors (laser-

related deviations, CCD-related deviations) and other unmeasurable errors. To reduce the disparity between the ideal simulated ONN model and the real physical ONN model, a physics-constrained ONN learning framework is constructed for image classification tasks. Specifically, we introduce a well-designed loss function to mitigate the laser-related deviations and a CCD adjustment strategy to reduce the CCD-related deviations. In addition, a 'physics-prior-based' error compensation network is proposed to manage other unmeasurable errors. The effectiveness of our approach is demonstrated through extensive experiments on the MNIST, Fashion MNIST and QuickDraw16 datasets.

Although we aim to reduce the gap between the ideal simulated ONN model and the real physical ONN model, it is important to note that our research primarily focuses on the image classification task. Specifically, the intensity gap loss function we propose may not be applicable to other AI tasks. However, we believe that the quantifiable error analysis and processing methods could serve as valuable references for other applications, as the two quantifiable errors—light source instability and exposure time mismatches—are inevitable in current optical systems. For future work, we plan to develop a more generalizable approach to address systematic errors in ONN systems and investigate its application in other AI tasks, such as image restoration and image segmentation.

## Acknowledgments and Disclosure of Funding

This work was supported by the Shanghai Outstanding Academic Leaders Plan (No. 20XD1402000) and the Scientific and Technological Innovation Funds of Shanghai Jiao Tong University. Additionally, it received partial support from the National Natural Science Foundation of China under grant U23B2011.

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

## NeurIPS paper checklis

