# OpenReview forum: "Physics-Constrained Comprehensive Optical Neural  Networks"
_NeurIPS.cc/2024/Conference — NeurIPS 2024 poster_

### Official Review · Reviewer_fa6W · 2024-07-06

**Soundness:** 3
**Presentation:** 3
**Contribution:** 3
**Rating:** 7
**Confidence:** 3

**Summary:**

The authors present a novel approach on using optical neural networks by integrating physical constrains into the training process. This work is motivated by the issues that ONN models present, the difference between the ideal, simulated system and the real, physical implementation — which significantly degrades the model performance.

The authors present a physics-constrained ONN that models and compensates for quantifiable physical errors (e.g., laser instability and exposure times mismatches in the camera). The authors conduct several classifications experiments on the framework, using MNIST, QuickDraw16 and Fashion MNIST; the results demonstrate the framework effectiveness with significant enhancement on the classification accuracy (e.g., from 61.6%, not using these corrections, to 93.5% when enabling the error compensation.

**Strengths:**

The authors present a learning method for ONNs which integrates quantifiable physical errors into the training process of the ONN. By doing this, the authors enhance the model capabilities in image classification, demonstrating high accuracy. This approach builds upon existing work in the field but delivers significantly advances on the classification accuracies for different datasets. This work improves the ONNs’ state of the art, enabling a more precise and robust optical computing applications.

The paper is well written and structured, and the level of detail is adequate for a broader audience.

**Weaknesses:**

1. The generalization of the method to broader applications, not only image classification, and its scalability remains unclear.


2. The performance depends on the accuracy of the physical models used to represent the system. This could limit the applicability of the framework in environments/situations where the physical modeling is challenging or impractical.


3. The authors discuss measurable physical errors — however, less insight is provided into the impact of unpredictable errors, which could be of significant importance when applying the framework to dynamic environments.

**Questions:**

1. the integration of physical constraints might introduce additional complexity in the training — which could potentially limit its use in more resource-constrained situations. How to the authors address that?

2. How does the model adapt to previously unseen physical environments? I am thinking, for example, as an implementation in autonomous vehicles or robotic systems.

**Limitations:**

The authors briefly mention the limitations of this study, focusing specifically on the demonstration only for classification tasks. However, questions such as the application of this method on dynamic, unpredictable environments, remains unclear.

---

> ### Author Rebuttal · Authors · 2024-08-06
>
> Thank you for your positive comments and valuable suggestions for improving the quality of our manuscript. As you rightly pointed out, our work is indeed motivated by the challenges presented by ONN models, specifically the differences between the ideal, simulated systems and their real, physical implementations. Furthermore, as you noted, our ultimate goal is to achieve more precise and robust optical computing applications. We have carefully considered your comments, and below are our responses to your feedback.
> 1：The generalization of the method to broader applications, not only image classification, and its scalability remains unclear.
>
> Thanks for your insightful question. As we mentioned in our conclusion. Although our loss design is tailored for classification tasks, the focus of our work is on compensating for various known and unknown errors in the physical system to achieve consistency between experimental and simulation results. Therefore, the framework of our method can still be applied to other tasks. We will explore the scalability of our approach to different tasks in future research.
>
> 2：The performance depends on the accuracy of the physical models used to represent the system. This could limit the applicability of the framework in environments/situations where the physical modeling is challenging or impractical.
>
> Your question is very practical. Our work has been validated for accuracy in the relatively simple 4f physical system. To achieve consistent results with more complex and challenging physical models, we would need to optimize and adjust the DNN architecture to adapt to these more complex systems, as well as integrate more quantitative physical information extracted from the physical system into the network training.
>
> 3：The authors discuss measurable physical errors — however, less insight is provided into the impact of unpredictable errors, which could be of significant importance when applying the framework to dynamic environments.
>
> Thank you for your valuable feedback. In our next steps, we will consider testing in dynamic environments.
> In addition to static and fixed unpredictable errors in experiments, some other errors can vary with changes in system parameters and often interact with each other. Modeling and analyzing their formation process is quite challenging. However, there are certain characteristic disturbances these errors cause in the experimental output images. Therefore, we chose to use DNNs to learn the impact of unpredictable errors by comparing ideal simulation images with actual experimental outputs, ultimately reducing their effect on experimental accuracy. For dynamic environments, we may need to further explore new DNN architectures and training methods based on the findings of this work.
>
> 4：the integration of physical constraints might introduce additional complexity in the training — which could potentially limit its use in more resource-constrained situations. How to the authors address that?
>
> The method proposed does add training burdens. However, it provides computational efficiency advantages compared to other error correction methods (as detailed in the "Convergence Rate and Model Architecture" section of the supplementary materials. By incorporating quantitative physical information, our DNN remains simple with fewer parameters, enhancing efficiency. Our focus is on model deployment and application; once trained, the model improves experimental results without altering the physical system. The inference process does not require additional computational resources due to the physical constraints introduced during training.
>
> 5：How does the model adapt to previously unseen physical environments? I am thinking, for example, as an implementation in autonomous vehicles or robotic systems.
>
> You’ve raised an excellent point that aligns with our ultimate goal. In automotive and robotics applications, issues such as slight changes in the relative positions of optical components, vibrations, and significant variations in laser intensity may arise. Since our current optical experiments are conducted in a controlled laboratory environment, further research is needed to address unpredictable errors in dynamic systems. We greatly appreciate your practical insight, as it will guide our future research and provide direction for the practical implementation of ONNs. We will continue to work towards this goal in future research, aiming to enhance ONN performance in previously unseen physical environments.
>
> 6. Limitations: The authors briefly mention the limitations of this study, focusing specifically on the demonstration only for classification tasks. However, questions such as the application of this method on dynamic, unpredictable environments, remains unclear.
>
> We will add the following paragraph to the paper to discuss the limitations mentioned by the reviewer:
> “Our method was developed and tested in a laboratory environment, where most errors are static and some are predictable. By incorporating predictable errors as physical priors into network training along with a relatively lightweight DNN architecture, we achieved consistency between simulation and experimental results. However, in dynamic and unpredictable environments, our proposed method would require more complex DNN architectures and new training approaches. We will work towards this direction in future research to develop ONNs that can operate stably and accurately under dynamic and unpredictable conditions.”
>
> We sincerely appreciate the valuable feedback and insightful comments you have provided. Your suggestions have been instrumental in refining our work and enhancing the quality of our manuscript. Thank you for your thoughtful review and for contributing to the improvement of our research.

---

> > ### Comment · Reviewer_fa6W · 2024-08-12
> >
> > Thank you for your clarifications. I don't have further comments. I have also read the rebuttal comment and I would like to keep my score.

---

### Official Review · Reviewer_W9o4 · 2024-07-08

**Soundness:** 3
**Presentation:** 3
**Contribution:** 2
**Rating:** 5
**Confidence:** 4

**Summary:**

In this paper, the author proposed the use of a 4f optical system to implement alternative neural networks due to the advantages of reduced latency and lower power consumption. The main challenges identified are optical misalignments caused by dispersion and imperfections in optical fiber fabrications.

**Strengths:**

1. **Motivation:** The motivation is clear, since errors from optical systems are unavoidable.
2. **Experimental Improvement:** The experiments showed a clear improvement, achieving approximately a 20% enhancement and presenting state-of-the-art ONN approaches on the MNIST dataset.

**Weaknesses:**

1. **Limited Experimentation:** The experiments were conducted only on MNIST or similar, similar challenging tasks, which can be effectively addressed even with basic approaches. The author might need to validate their approach on more complex datasets. A simple single-layer fully connected (FC) network could handle MNIST, so the robustness of the proposed method on more challenging tasks should be explored.
2. **Broader Impact:** The author should emphasize the benefits of optical neural networks (ONNs) for a broader audience, such as comparing power consumption more comprehensively.
3. **Previous Work:** Some important previous work, such as from the "meta surface ONN" field, might be missing. Notable references include or discussed:
   - [1] Photonic Advantage of Optical Encoders
   - [2] Compressed Meta-Optical Encoder for Image Classification
   - [3] The Physics of Optical Computing
   - [4] Optical convolutional neural network with atomic nonlinearity

**Questions:**

1. One drawback of the 4f approach is that it requires three elements (two lenses and a spatial filter), resulting in a bulky optical system with a greater propensity for misalignments compared to single-element optical systems. Therefore, The author might need to explain why use the 4f system, and how does it compare with meta surface approaches?

2. It is also unclear how the author handles the nonlinearity issue. The author only mentioned that they use a linear ONN, but dropping nonlinearity from NNs is non-trivial. It might be better to explain this gap.

**Limitations:**

1. The experiments were conducted only on the MNIST level dataset. The 4f system might not perform as well on more challenging datasets, such as CIFAR or ImageNet. While I don't think the author needs to conduct these additional experiments, some discussions would be great.

---

> ### Author Rebuttal · Authors · 2024-08-06
>
> Thank you for your valuable feedback and for acknowledging the motivation and improvements in our study. Our work addresses errors in practical optical neural networks, enhances experimental performance, and aims to advance reliable optical computing. We will reply your questions and suggestions below.
>
> 1: Limited Experimentation: The experiments were conducted only on MNIST or similar, similar challenging tasks, which can be effectively addressed even with basic approaches. The author might need to validate their approach on more complex datasets. A simple single-layer fully connected (FC) network could handle MNIST, so the robustness of the proposed method on more challenging tasks should be explored.
>
> Thank you for your question about the robustness of our method. Our approach can handle more challenging datasets like CIFAR-100. To do this, we simply replace the dataset loaded onto the DMD with CIFAR-100 images. The initial classification accuracy with CIFAR-100 is 30%, which improves to 57% after optimization using our method. However, since current research ("Dual Adaptive Training of Photonic Neural Networks" published in Nature Machine Intelligence, and "Deep Physical Neural Networks Trained with Backpropagation" published in Nature.) on spatial 2D light-based ONNs predominantly uses the datasets mentioned in the paper, we did not include the CIFAR-100 classification results in the main text due to the lack of comparative benchmarks.
> Moreover, the core contribution of this paper is addressing the inconsistency between simulation and actual optical neural networks, rather than just developing a high-performing ONN. For instance, experimental accuracy for MNIST was 61.6% without our method, but improved to 96.5% with it. Similarly, accuracies for CIFAR-100 and Quickdraw16 increased from 30% and 63% to 57% and 85.7%, respectively. These results validate the effectiveness and robustness of our method in correcting experimental errors.
>
> 2：One drawback of the 4f approach is that it requires three elements (two lenses and a spatial filter), resulting in a bulky optical system with a greater propensity for misalignments compared to single-element optical systems. Therefore, the author might need to explain why use the 4f system, and how does it compare with meta surface approaches?
>
> Thank you for pointing out the drawback of the 4f approach. We agree that it is more prone to misalignments than single-element systems, and its bulkiness complicates error correction. However, since the errors analyzed are common to optical systems, our method can also be effectively applied to single-element systems, where misalignment errors are easier to manage.
> We agree that the meta surface approach is better for spatially structured light computing due to its industrial potential and benefits for miniaturizing ONNs. Our other team is already working on meta surface-based ONNs. We used the 4F system here because it's a classic setup that allows us to validate our method in the laboratory, but our approach is also applicable to meta surface-based ONNs.
>
> 3：It is also unclear how the author handles the nonlinearity issue. The author only mentioned that they use a linear ONN, but dropping nonlinearity from NNs is non-trivial. It might be better to explain this gap.
>
> Your comments on nonlinearity are crucial. Our ONN includes a nonlinear activation function at the camera's receiving surface, performing a squared transformation on the light field matrix to output its squared magnitude. This operation is recognized as nonlinear in the literature, such as "Optronic Convolutional Neural Networks of Multi-Layers with Different Functions Executed in Optics for Image Classification" in Optics Express. In both our simulations and optical experiments, we incorporated this nonlinear operation. However, due to noise, this nonlinearity did not improve experimental accuracy. Nonetheless, our method can also ensure consistency between simulation and experimental results, ultimately enhancing accuracy.
>
> 4. Broader Impact: The author should emphasize the benefits of optical neural networks (ONNs) for a broader audience, such as comparing power consumption more comprehensively. Previous Work: Some important previous work, such as from the "meta surface ONN" field, might be missing. Notable references include or discussed.
>
> Thank you for your suggestion. The advantages of optical neural networks regarding energy consumption and processing speed are mentioned in the Introduction and discussed in the referenced articles. We will add further discussion to the manuscript to comprehensively cover these advantages.
> “ONNs offer significant advantages over traditional electronic neural networks, including high-speed computations and low latency due to massive parallel processing with light. This efficiency reduces power consumption, handles complex tasks effectively, and is scalable for compact systems. Additionally, light transmission minimizes electrical interference and heat dissipation, enhancing performance and reliability.”
>
> 5. Limitations:
> The experiments were conducted only on the MNIST level dataset. The 4f system might not perform as well on more challenging datasets, such as CIFAR or ImageNet. While I don't think the author needs to conduct these additional experiments, some discussions would be great.
>
> Thank you for your concern. This issue is mentioned in our reply to Question 1, Our work can also handle more challenging datasets as well, such as the CIFAR-100 dataset, which indicates that the method proposed in the paper exhibits a degree of generalization capability.
>
> Finally, we would like to express our gratitude for your valuable feedback. In our future work, we aim to contribute to the practical implementation of metasurfaces for optical neural networks.

---

> > ### Comment · Reviewer_W9o4 · 2024-08-12
> >
> > Since the author addressed most of my concerns and added further explanation on CIFAR-10, I would increase the score. I suggest that the author include a brief comparison with the 4F and Metasurface approaches.

---

### Official Review · Reviewer_M382 · 2024-07-13

**Soundness:** 2
**Presentation:** 2
**Contribution:** 1
**Rating:** 3
**Confidence:** 5

**Summary:**

This paper proposes to use two analytical error terms to model laser jitter noise and camera exposure time error, and use a DNN-based error compensation network to correct unknown noises for higher accuracy after deployment.

**Strengths:**

It has thorough ablation study on the effectiveness of the error compensation terms and DNN compensation model and have experimental results to validate the effectiveness.

**Weaknesses:**

Weaknesses:
1.	Novelty is quite limited. Using physical or analytical error modeling in training for error compensation is common practice. Using a neural network to fit unknown errors to emulate the response of real hardware is also widely used in analog ONNs. What novel techniques are proposed here, especially for NeurIPS, which is the main ML contribution? Claiming “analytical noise modeling in optimization” as physics-informed neural network will not make it a novel contribution.

2.	The parallel residual DNN compensation network on digital hardware significantly reduces the system throughput and efficiency. What are the main advantages of such a system compared to analog AI hardware or digital accelerators?

3.	The experiments are performed in very small examples. For ONNs submitted to NeurIPS, CIFAR100 or TinyImageNet are required.

4.	It does not compare the efficiency and robustness with extensive literature on robustness-driven hardware/software co-design methods or on-chip training, hybrid training methods. It only compares to direct deployment.

**Questions:**

Weaknesses:
1.	Novelty is quite limited. Using physical or analytical error modeling in training for error compensation is common practice. Using a neural network to fit unknown errors to emulate the response of real hardware is also widely used in analog ONNs. What novel techniques are proposed here, especially for NeurIPS, which is the main ML contribution? Claiming “analytical noise modeling in optimization” as physics-informed neural network will not make it a novel contribution.

2.	The parallel residual DNN compensation network on digital hardware significantly reduces the system throughput and efficiency. What are the main advantages of such a system compared to analog AI hardware or digital accelerators?

3.	The experiments are performed in very small examples. For ONNs submitted to NeurIPS, CIFAR100 or TinyImageNet are required.

4.	It does not compare the efficiency and robustness with extensive literature on robustness-driven hardware/software co-design methods or on-chip training, hybrid training methods. It only compares to direct deployment.

---

> ### Author Rebuttal · Authors · 2024-08-06
>
> I am very grateful to your comments for the manuscript. Your questions were answered below.
>
> 1: Novelty is quite limited. (1) Using physical or analytical error modeling in training for error compensation is common practice. (2) Using a neural network to fit unknown errors to emulate the response of real hardware is also widely used in analog ONNs.
>
> The purpose of this paper is not to emulate the response of actual hardware for building simulation-based ONN models. Instead, we aim to use deep neural networks to ensure that the actual optical path accurately replicates the results of the simulation model, ultimately achieving consistency between simulation and experimental results.
>
> Regarding point (1), while error compensation training based on physical error model analysis is widely used in Physics-Informed Neural Networks (PINNs), this work is the first work to conduct large-scale physical experimental analysis of light source instability and exposure time mismatches in optical neural networks (ONNs). These issues are common in optical setups and significantly impact the final experimental accuracy. We have proposed corresponding solutions to address the issue. As Reviewer 8n42 mentioned, our proposed strategies have significantly narrowed the gap between simulated and physical ONNs. Our research thus provides a pathway to more accurate and reliable optical computing applications. Moreover, the physical experimental results we have documented can serve as a valuable reference for other researchers engaged in innovations in optical networks.
>
> Regarding point (2), unmeasurable errors in optical systems are categorized into fixed and coupling errors (analyzed in Section 3.3 of the main text). While fixed errors can be addressed using DNNs, our approach updates optical system parameters each epoch, causing changes in coupling errors. We synchronize DNN updates with these parameters, allowing the DNN to correct both fixed and coupling errors more effectively. This synchronization has significantly improved experimental ONN performance. As noted by Reviewer W9o4, our method achieves state-of-the-art results on the MNIST dataset. Our method represents a significant innovation, offering unique advantages over existing approaches and demonstrating its potential for laying the foundation for the practical development of ONNs.
>
> 2: The parallel residual DNN compensation network on digital hardware significantly reduces the system throughput and efficiency. What are the main advantages of such a system compared to analog AI hardware or digital accelerators?
>
> As noted, our approach uses residual connections between the ideal ONN model and the DNN to ensure consistency between the simulation model and the actual optical path. The DNN's ~5,000 parameters, compared to the ONN's 360,000(as detailed in the supplementary materials under Model Architecture), minimally impact system efficiency while improving consistency between experimental and simulation results.
>
> Additionally, the DNN affects only the training complexity, not the inference process. In practical engineering applications, we are more concerned with the inference process rather than the training process. During inference, ONNs offer advantages such as low power consumption, high parallelism, and strong resistance to interference compared to analog AI hardware or digital accelerators. These advantages are highlighted in the papers “Deep Physical Neural Networks Trained with Backpropagation” published in Nature and so on.
>
> 3: The experiments are performed in very small examples. For ONNs submitted to NeurIPS, CIFAR100 or TinyImageNet are required.
>
> For the CIFAR-100 dataset, our system was adapted by replacing the DMD images with CIFAR-100 images, maintaining the training process as described. For CIFAR-100, we used only one channel of grayscale images for training and forward experiments, achieving an improvement in classification accuracy from an initial 30% to 57%, further demonstrating the effectiveness of our method. Due to a lack of comparative results, this was not included in the main text. Current ONNs using 2D spatial structured light primarily utilize the datasets mentioned in our paper, as noted in the aforementioned articles and "Hybrid Training of Optical Neural Networks" in Optica.
>
> 4: It does not compare the efficiency and robustness with extensive literature on robustness-driven hardware/software co-design methods or on-chip training, hybrid training methods. It only compares to direct deployment.
>
> Table 1 compares our classification performance with methods from "Dual Adaptive Training of Photonic Neural Networks" in Nature Machine Intelligence and "Deep Physical Neural Networks Trained with Backpropagation" in Nature. Section B of the supplementary materials details the convergence rate, showing that our method achieves convergence within 5 epochs despite the lower precision of initial experimental deployment. These results underscore our advantages in both accuracy and training efficiency.
>
> In Table 1, PAT and DAT are hybrid training methods. Our results show that our method outperforms these in both training efficiency and final experimental accuracy. On-chip learning, which involves backpropagating optical signals through hardware, introduces significant experimental errors and is thus highly challenging. Most researchers use optical forward inference combined with loss function computation and backpropagation in computer systems, as seen in the papers "Deep Physical Neural Networks Trained with Backpropagation" in Nature and "Backpropagation-Free Training of Deep Physical Neural Networks" in Science. Therefore, we did not compare our method with on-chip learning approaches.
>
> Finally, we sincerely thank you for your valuable feedback on our paper. We will address the issues you raised through further in-depth research.

---

> > ### Author Response · Authors · 2024-08-13
> >
> > Dear Reviewer M382,
> > We understand that you are very busy and maybe working on our rebuttal, and we are sorry for the misunderstanding brought to you due to the missing of some important details in our submitted manuscript. As can be seen that all the other three reviewers have provided comments on our rebuttal and give positive feedback stressing that our response have addressed nearly all their concerns, and acknowledged our contribution again. Therefore, we want to kindly remind you that we are still waiting for your valuable comments on our rebuttal to further know what issues are remained from your aspect, so that we can further improve our paper. Thank you very much for your effort on our paper, and look forward to hearing from your response as soon as possible.

---

> > ### Comment · Reviewer_M382 · 2024-08-14
> > **Thanks for the responses.**
> >
> > Thanks for the responses. Adding neural networks to project simulated results to match the experimental results to get a more accurate model for backpropagation is a standard technique in analog NN calibration that many people used years ago by default. The novelty is not well justified. Experimental results are important; however, the engineering effort paid is not a machine learning contribution, which will be given more value if submitted to Optica or Nature journals. I will keep my original score.

---

> ### Author Response · Authors · 2024-08-14
>
> We acknowledge that using neural networks for error compensation is a common technique in analog NN calibration. However, the errors present in physical systems are often too complex to be fully addressed by simple neural networks. As discussed in our experiments, even with the aid of a compensation DNN, the classification accuracy on the MNIST dataset reached only 93.5%. By innovatively integrating quantitative physical information into the model design and training, guide the design of the neural network and narrow the optimization scope of key parameters using the physical model's transmission process and known critical parameters. we were able to improve the accuracy from 93.5% to 96.5%, demonstrating the effectiveness of our approach.
>
> Therefore, we argue that our contributions extend beyond the solid engineering efforts; they include original observations and innovative ideas specifically designed to bridge the gap between simulated and physical ONNs, thus laying the foundation for the practical development of ONNs.
>
> What’s more, for the research of error compensation and backpropagation algorithms for optical neural networks, which is a multidisciplinary topic covering optic、computer science、artificial intelligence, the related papers are accepted/published at different venues. Some are published at top-tier journals, e.g., Nature, Science etc., which are deemed as comprehensive journals, and some are published at more specific journals, e.g., Nature Machine Intelligence, Nature Computational Science etc. Therefore, we believe our manuscript is well suited to and adequately match the coverage of NeurIPS, which is a broad and inclusive platform that welcomes contributions across all areas of artificial intelligence, including the application and integration of engineering efforts that advance the field. We believe that our original research, which addresses the gap between simulation and experimental results in ONNs, holds significant value for the AI community. The impact of NeurIPS is comparable to that of high-impact journals, and we are confident that our work can spark interest and inspire further research across the fields of AI and machine learning.

---

### Official Review · Reviewer_8n42 · 2024-07-15

**Soundness:** 3
**Presentation:** 2
**Contribution:** 3
**Rating:** 7
**Confidence:** 3

**Summary:**

This work presents a physics-constrained framework to improve optical neural networks for image classification. The study identifies and reducing two key errors: light source instability and exposure time mismatches.The authors introduce a physics-constrained ONN learning framework that includes a specialized loss function to reduce light fluctuations, a CCD adjustment strategy to handle exposure time mismatches, and a physics-prior-based error compensation network to manage other systematic errors. This approach significantly enhances classification accuracy on various datasets, achieving 96.5% on MNIST, 85.7% on QuickDraw16, and 77.5% on Fashion MNIST, compared to lower baseline performances.

**Strengths:**

Originality: The paper introduces a novel approach by implementing a physics-constrained learning framework to enhance the performance of optical neural networks (ONNs). This framework specifically targets physical errors like light source instability and exposure time mismatches, providing unique solutions to these problems.
Quality: The authors identify and quantify key sources of error in physical ONNs and offer practical solutions to mitigate these errors. The substantial improvements in classification accuracy across several datasets underscore the robustness and effectiveness of the framework.
Clarity:The methodology is explained in detail, with clear descriptions of the error sources, proposed solutions, and the experimental setup. Diagrams and equations clear illustrate the points.
Significance: By narrowing the gap between simulated and physical ONNs, the research provides a way to more accurate and reliable optical computing applications.

**Weaknesses:**

Providing an analysis of the computational costs associated with the proposed framework, including comparisons of training and inference times.
Including a more in-depth analysis of how the proposed methods differ fundamentally from other approaches would clarify their unique advantages.

**Questions:**

How well does your framework generalize to datasets with different characteristics (e.g., noise levels, resolutions, and complexity)?

How did you determine the parameters for the loss function?

**Limitations:**

The authors have identified key limitations, particularly their focus on image classification tasks and the specific applicability of their intensity gap loss function. I think autohors should also discuss how the framework might perform under real-world conditions in this work.

---

> ### Author Rebuttal · Authors · 2024-08-06
>
> We greatly appreciate your thorough review and the positive feedback on our work. As you mentioned, the core aim of our study is to narrow the gap between simulated and physical ONNs and to provide a pathway for more accurate and reliable optical computing applications. Your description is a very precise summary of our paper.
>
> 1: (1) Providing an analysis of the computational costs associated with the proposed framework, including comparisons of training and inference times. (2) Including a more in-depth analysis of how the proposed methods differ fundamentally from other approaches would clarify their unique advantages.
>
> Regarding (1), during the training of the ONN, we train based on the transmission equations of light in a 4f system, achieving end-to-end mapping through the adjustment of the system's tunable parameters. The DNN used is a lightweight fully connected structure, which converges quickly. The only limitation on training efficiency is the acquisition of experimental images. To address this, we developed a synchronized trigger control unit to ensure that image refresh rates in the DMD (Digital Micromirror Device) are synchronized with camera image capture. The DMD refresh rate is typically very high, capable of refreshing an image every 10 microseconds, but industrial cameras often struggle to match this speed. Consequently, training and forward inference efficiency depend on the camera frame rate; using a higher frame rate camera can improve training efficiency and accelerate inference times.
>
> Regarding (2), unlike other methods, we are the first to incorporate quantitative physical information into network training. By integrating this quantitative physical data into the model training, we are able to reduce model complexity of error compensation networks and enhance convergence efficiency as discussed in Convergence Rate section of supplemental material.
>
> 2：（1）How well does your framework generalize to datasets with different characteristics (e.g., noise levels, resolutions, and complexity)?（2）How did you determine the parameters for the loss function?
>
> Our experiments were conducted on three datasets with different characteristics: MNIST, Fashion-MNIST, and QuickDraw16. Additionally, we also processed the CIFAR-100 dataset. The features extracted for classification from these datasets and the number of categories differ significantly, and there is substantial variation in the brightness and contrast among the datasets. Additionally, the optical paths used in practice are highly sensitive to variations in brightness and contrast. For the MNIST and CIFAR-100 datasets, our method improved accuracy from 61.6% and 30% to 96.5% and 57%, respectively. Ablation studies across these datasets demonstrate that our method consistently aligns simulation and experimental results and significantly enhances experimental accuracy for datasets with varying features. Regarding the noise levels in the datasets, due to time constraints, we plan to incorporate noise into the data in upcoming experiments to further assess the generalization capability of our method.
>
> The parameters for the loss function are determined based on physical prior information. For example, due to factors such as laser jitter, camera dark current, and other irregular noises, the grayscale values of the same region in an image are not constant but fluctuate over time. We record the range of grayscale value fluctuations and use this information to design the relevant parameters in the loss function.
>
> 3：Limitations：The authors have identified key limitations, particularly their focus on image classification tasks and the specific applicability of their intensity gap loss function. I think authors should also discuss how the framework might perform under real-world conditions in this work.
>
> The issues you raised are highly practical. As shown in the Experimental Setup section of the supplementary materials, the actual optical layout of the 4f system indeed presents challenges such as large volume, numerous optical components, and difficulty in optical alignment. However, our proposed method addresses common errors in optical systems and ensures consistency between experimental and simulation results, significantly improving the final experimental accuracy even in error-prone 4f systems. Therefore, we believe that our method can be effectively applied to single-element systems, such as metasurface-based ONNs, which are more suited for real-world conditions.
>
> We sincerely thank you for your valuable feedback and thoughtful comments. Your insights have greatly contributed to improving the quality of our manuscript. We appreciate the time and effort you have dedicated to reviewing our work.

---

> > ### Comment · Reviewer_8n42 · 2024-08-12
> >
> > Thank you for your thoughtful and detailed response. I appreciate the explanations you’ve provided, particularly on the computational aspects and the unique approach of incorporating quantitative physical information into your model. Your insights into how you’ve managed synchronization challenges and used a lightweight DNN structure are very helpful.
> > I’m also reassured by your discussion on generalizing the framework across different datasets and your clear, well-grounded approach to setting loss function parameters. Your acknowledgment of the challenges in real-world applications and how your method could be adapted to more practical systems like metasurface-based ONNs shows a realistic and forward-thinking perspective.Overall, your responses have addressed my concerns well and provided a clearer understanding of your work.

---

### Author Rebuttal · Authors · 2024-08-06

We sincerely thank all reviewers for their thoughtful comments. We appreciate their findings that our work focuses on narrowing the gap between simulated and physical optical neural networks(8n42) by addressing the inevitable errors encountered during the physical realization of these networks(W9o4). Consequently, our approach has significantly enhanced the performance of experimental optical neural networks(W9o4，8n42，fa6W), achieving state-of-the-art classification accuracy across MNIST dataset(W9o4). This advancement not only demonstrates the effectiveness of our method but also provides a pathway to more accurate and reliable optical computing applications(8n42，fa6W). Large-scale physical ablation experiments also demonstrate the efficacy of our proposed method in compensating for system errors and enhancing the consistency between experimental and simulated results.(M382,8n42). We will incorporate all suggestions. We sincerely hope this will help reviewers finalize their evaluations.

M382,W9o4. Contributions of our paper.Our work is the first work to conduct large-scale physical experimental analysis of light source instability and exposure time mismatches in optical neural networks (ONNs). These issues are common in optical setups and significantly impact the final experimental accuracy. Moreover, the core contribution of this paper is not merely the development of a high-performing optical neural network, but rather the resolution of the widespread issue of inconsistency between simulation systems and actual optical neural networks. We conducted extensive ablation experiments in a 4f system with numerous sources of error, which demonstrate both the efficiency and accuracy of our approach.

M382,W9o4,8n42. Datasets used in our paper. Our work can indeed handle more challenging datasets as well, such as the CIFAR-100 dataset. To do this, we simply replace the MNIST or Fashion-MNIST dataset loaded onto the DMD with CIFAR-100 images. Due to the lack of comparative results with other studies, we did not include this result in the main text. Additionally, for current ONNs based on 2D spatial structured light, researchers（The aforementioned articles, along with the paper 'Hybrid Training of Optical Neural Networks' published in Optica） primarily use the datasets mentioned in our paper. Our experiments were conducted on three datasets with different characteristics, and ablation studies across these datasets demonstrate that our method consistently aligns simulation and experimental results and significantly enhances experimental accuracy for datasets with varying features.

M382, W9o4. The benefits of optical neural networks. Optical neural networks (ONNs) offer several compelling advantages over traditional electronic neural networks. They can perform massive parallel processing with light, leading to extremely high-speed computations and low latency compared to analog AI hardware or digital accelerators.

fa6W,M382. The training efficiency of our approach. Unlike other methods, we are the first to incorporate quantitative physical information into network training. By integrating this quantitative physical data into the model training, we are able to reduce model complexity and enhance convergence efficiency. Hence, compared to the 360,000 parameters of the ONN, the DNN's approximately 5,000 trainable parameters (as detailed in the supplementary materials under Model Architecture) have minimal impact on training efficiency while significantly improving the consistency between experimental and simulation results. Moreover, the addition of DNN only affects the complexity and computational burden of the training process, without impacting the inference process of the physical ONN. In practical engineering applications, we are more concerned with the inference process rather than the training process.

8n42,fa6W. The performance of our work in practical environments. As shown in the Experimental Setup section of the supplementary materials, the actual optical layout of the 4f system indeed presents challenges such as large volume, numerous optical components, and difficulty in optical alignment. However, our proposed method addresses common errors in optical systems and ensures consistency between experimental and simulation results, significantly improving the final experimental accuracy even in error-prone 4f systems. Therefore, I believe that our method can be effectively applied to single-element systems, such as meta surface-based ONNs, and perform well under real-world conditions. However, in dynamic and unpredictable environments, our proposed method would require more complex DNN architectures and new training approaches. We will work towards this direction in future research to develop ONNs that can operate stably and accurately under dynamic and unpredictable conditions.

M382, W9o4, fa6W. Some other questions. (i) Table 1 in the main text presents a comparative analysis of our classification performance against the hybrid training methods, which highlights our advantages in both accuracy. (ii) The bulkiness of 4f’s structure indeed makes it more challenging to address system errors. Given that the types of errors analyzed in this work are common to optical systems, we believe our method can be effectively applied to single-element optical systems as well, where misalignment errors are easier to handle. And we also agree that the meta surface approach is a better choice for realizing spatially structured light computing due to its significant potential for industrial applications and its role in miniaturizing and lightening ONNs. We have another team that has just started working on meta surface-based ONNs. (iii) We will include a section in the paper discussing the impact of unpredictable errors on experimental results. (iv) We will polish the writing in the revised manuscript.

---

### Decision · Program_Chairs · 2024-09-25

**Decision:**

Accept (poster)

**Comment:**

All reviewers agree on the originality of the problem addressed in the paper and the quality of the experimental validations.

Reviewers M382, W9o4, fa6W ask for more challenging data experiments and authors provide further results for CIFAR10 and show that their technique can improve in that scenario as well.

Reviewer M382 is not satisfied with the novelty of the proposed DNN-based error calibration method and considers this to be a straightforward engineering solution. Authors reply to the reviewer by showing how their mathematical modeling of noise in the training objective can boost the performance on top of the DNN-based approach.